# Type 2 Diabetes Mellitus in Low- and Middle-Income Countries: The Significant Impact of Short-Chain Fatty Acids and Their Quantification

**DOI:** 10.3390/diagnostics14151636

**Published:** 2024-07-30

**Authors:** Scelo Khumalo, Zamathombeni Duma, Lizette Bekker, Koketso Nkoana, Sara Mosima Pheeha

**Affiliations:** 1Department of Chemical Pathology, Sefako Makgatho Health Sciences University, Molotlegi Street, Ga-Rankuwa Zone 1, Ga-Rankuwa 0208, South Africa; zamathombeni.duma@smu.ac.za (Z.D.); lizette.bekker@gmail.com (L.B.); koketso.nkoana@smu.ac.za (K.N.); 2National Health Laboratory Service, Dr George Mukhari Academic Hospital, Pretoria 0208, South Africa; pheeha.sm@gmail.com; 3Division of Epidemiology and Biostatistics, Faculty of Medicine and Health Sciences, Stellenbosch University, Cape Town 7500, South Africa

**Keywords:** short-chain fatty acids, type 2 diabetes mellitus, LMICs

## Abstract

Globally, type 2 diabetes mellitus (T2DM) is a major threat to the public’s health, particularly in low- and middle-income countries (LMICs). The production of short-chain fatty acids (SCFAs) by the gut microbiota has been reported to have the potential to reduce the prevalence of T2DM, particularly in LMICs where the disease is becoming more common. Dietary fibers are the primary source of SCFAs; they can be categorized as soluble (such as pectin and inulin) or insoluble (such as resistant starches). Increased consumption of processed carbohydrates, in conjunction with insufficient consumption of dietary fiber, has been identified as a significant risk factor for type 2 diabetes (T2DM). However, there are still controversies over the therapeutic advantages of SCFAs on human glucose homeostasis, due to a lack of studies in this area. Hence, a few questions need to be addressed to gain a better understanding of the beneficial link between SCFAs and glucose metabolism. These include the following: What are the biochemistry and biosynthesis of SCFAs? What role do SCFAs play in the pathology of T2DM? What is the most cost-effective strategy that can be employed by LMICs with limited laboratory resources to enhance their understanding of the beneficial function of SCFAs in patients with T2DM? To address the aforementioned questions, this paper aims to review the existing literature on the protective roles that SCFAs have in patients with T2DM. This paper further discusses possible cost-effective and accurate strategies to quantify SCFAs, which may be recommended for implementation by LMICs as preventive measures to lower the risk of T2DM.

## 1. Introduction

Diabetes mellitus (DM) is a chronic metabolic disease characterized by increased blood glucose levels (hyperglycemia), resulting from either the deficiency or action of insulin [1]. If left untreated for some time, the disease may damage various body organs, such as the kidneys, eyes, vasculature, nerves, and the cardiovascular system [1]. More than 90% of DM cases are Type 2 diabetes mellitus (T2DM) globally; however, this proportion varies widely among countries [1]. T2DM affects an estimated 530 million adults globally, and its prevalence is gradually increasing [2,3]. Research has shown that T2DM is the primary cause of death and financial loss. T2DM is currently recognized as a serious threat to the world’s economic growth [1]. The reason for this is that T2DM is associated with a high financial burden on the health system due to its chronic nature and associated complications [3]. What raises serious concerns is the fact that 80% of the 530 million adult diabetics worldwide reside in low- and middle-income countries (LMICs). This statistic highlights the significant disparities in healthcare access, resources, and awareness across the world [3]. Therefore, without concerted efforts to address the root causes of diabetes and improve healthcare systems globally, the burden of the disease will continue to disproportionately affect the vulnerable populations in LMICs.

The rise in the prevalence of T2DM is mostly due to lifestyle and environmental changes that are associated with sedentary behaviors and poor diets that lack nutritional quality [4]. Avoiding unhealthy habits and incorporating ideal practices, such as physical activity and eating healthy nutritious food (containing adequate dietary fiber) may reduce the risk of T2DM, disease severity, or even reverse T2DM [5]. Several studies have consistently shown that dietary fiber consumption reduces the risk of T2DM [6]. Other studies have linked health benefits associated with the production of short-chain fatty acids (SCFAs) by the gut microorganisms from dietary fiber, as well as their subsequent absorption and use by the host [7]. Addressing dietary patterns, particularly the imbalance between processed carbohydrates and dietary fiber intake, is essential for both preventing and managing T2DM. Promoting diets rich in whole grains and minimally processed foods that are high in dietary fiber is proposed as an effective strategy for reducing the risk of T2DM and enhancing overall metabolic health [8].

Several studies have suggested that the disruption of short-chain fatty acid synthesis seen in T2DM patients is caused by pathogenic bacteria that are unable to produce SCFAs. These alterations may contribute to metabolic dysfunction, inflammation, and insulin resistance, which are observed in T2DM [9]. That is why there has been a growing interest in exploring SCFAs and gut microbiota as potential targets for T2DM management and understanding their role in disease pathogenesis [9]. Measuring SCFA levels in biological samples, such as feces or blood, may provide insights into the metabolic status and gut health of individuals with T2DM [10]. Interestingly, a large number of high-income Western countries have conducted research measuring SCFAs in T2DM patients’ blood and fecal samples; however, few studies have been carried out in LMICs, where T2DM prevalence is currently rising rapidly [10]. The lack of studies quantifying short-chain fatty acids (SCFAs) in both blood and fecal samples from T2DM patients in LMICs represents a gap in our understanding of the disease’s pathogenesis and management in these regions [10]. The quantitative measurement of SCFAs in human biological samples from LMICs is essential for advancing our understanding of T2DM pathogenesis and developing context-specific interventions to combat the disease in these regions [11]. Closing the research gap in SCFA studies between high-income Western countries and LMICs is crucial for promoting global health equity in T2DM research and care [11]. Therefore, the present review discusses an overview of SCFAs and their advantageous effects on health and diseases, especially T2DM, based on previous literature. The review will further outline a highly sensitive and cost-effective strategy for accurately quantifying SCFAs in human biological samples. This strategy may be beneficial for LMICs with limited laboratory resources to assess the levels of SCFAs and glucose metabolism in patients with T2DM.

## 2. Literature Search for the Current Review Study

The flow diagram (Figure 1) illustrates the strategy that was used to search the literature for the current narrative review.

## 3. The Biochemistry of Short-Chain Fatty Acids

Short-chain fatty acids are unsaturated fats with fewer than six carbon atoms that are produced in the colon and cecum by stomach microbes, as they digest dietary fibers, sugars (resistant starch), and other phytonutrients [12]. SCFAs are mainly produced through carbohydrate fermentation; however, they can also be produced through protein and amino acid breakdown [13]. They are classified into two groups: straight-chain SCFAs, produced in the gut, which primarily consist of acetic, propionic, and butyric acids, which are absorbed in various quantities from the large intestine, with some being absorbed most significantly locally and systematically, and formic and valeric acid [7]. The three major SCFAs account for 95% of all SCFAs and are found in the colon and stool in an approximate molar ratio of acetic acid (60%)/propionic acid (20%)/butyric acid (20%) [13]. The second group of SCFAs consists of three branched-chain SCFAs, known as isobutyric acid, 2-methylbutyric acid, and isovaleric acid, which are mostly produced by the breakdown of branched-chain amino acids such as valine, leucine, and isoleucine [7]. Despite being produced in smaller amounts, they have been identified as markers of protein fermentation, a process that simultaneously generates other fermentation products that are potentially harmful to the colon’s epithelium [14]. The majority of SCFAs are absorbed in the cecum and colon via protonated SCFA diffusion and anion exchange, with only 5–10% being excreted in the feces [13].

## 4. The Biosynthesis of Short-Chain Fatty Acids

The biosynthesis of the three main SCFAs involves distinct pathways: the Wood–Ljungdahl pathway for acetate, the combination of two acetate molecules for butyrate, and the acrylate, succinate, and propanediol pathways for propionate [15], as depicted in Figure 2. The gut microbiota breaks down nondigestible carbohydrates into monosaccharides, which are then fermented in the gut’s anaerobic environment [15]. The Embden–Meyerhof–Parnas pathway (glycolysis, for six-carbon sugars) and the Pentose–phosphate pathway (for five-carbon sugars) are the two major bacterial metabolic routes involved in the conversion of monosaccharides to phosphoenolpyruvate (PEP) [13]. PEP is then converted into fermentation products, which are SCFAs, through a series of biochemical reactions. With the formation of H_2_ and CO_2_, a significant portion of the pyruvate derived from PEP is converted into acetyl-CoA. Acetate is produced in two ways: through the hydrolysis of acetyl-CoA or via the Wood–Ljungdahl pathway, in which CO_2_ is reduced to CO and then converted with a methyl group and CoASH to acetyl-CoA [13].

Regarding propionate formation, nicotinamide adenine dinucleotide hydrogen (NADH) is a primitive anaerobic electron chain that oxidizes monosaccharides to PEP [15]. It begins with PEP carboxylation, and the resulting oxaloacetate is reduced to malate before being oxidized to fumarate. Then, via a simple electron-transfer chain between NADH and fumarate, fumarate accepts electrons from NADH. This chain involves two enzymes: NADH dehydrogenase and fumarate reductase. The protons are moved across the cell membrane by NADH dehydrogenase and used to produce ATP. When the partial pressure of CO_2_ is low, succinate, the product of fumarate reductase, is converted into methylmalonate, which is cleaved into propionate and CO_2_. Propionate can also be formed by reducing lactate to propionate, a process known as the acrylate pathway [15,16,17]. Finally, butyrate is produced by the condensation of two acetyl-CoA molecules to form acetoacetyl-CoA, which is then reduced to butyryl-CoA. The last step in the synthesis of butyrate from butyryl-CoA, involving two distinct methods: either the phospho-butyrate and butyrate kinase pathways or the butyryl-CoA: the acetate CoA–transferase route [13,16].

## 5. Several Major Gut Microbiota Involved in the Production of SCFAs

Short-chain fatty acids (SCFAs) are produced by several bacterial families in the gut and play significant roles in various physiological processes. Probiotics such as *Lactobacillus* and *Bifidobacterium* have been shown to improve lipid metabolism in the liver by increasing SCFA production [18]. According to Chambers et al. (2018), *Akkermansia muciniphila*, a probiotic that has the potential to lower the risk of metabolic syndrome, converts dietary fiber into SCFAs [19]. Furthermore, in both humans and animals, *Akkermansia muciniphila* can reverse metabolic dysbiosis and improve insulin resistance caused by antibiotics or a high-fat diet [20,21].

Acetate is produced by the gut microbiota, which includes *Blautia hydrogenotrophica*, *Bifidobacteria*, *Lactobacilli*, *Clostridium* species, and *Streptococcus* species. The genera Bifidobacteria and Lactobacilli [22], as well as other bacteria, including *Clostridium* spp. and *Streptococcus* spp., produce acetate [22,23]. Most of these gut bacteria use the acetyl-CoA pathway to convert pyruvate into acetate [24]. Interestingly, although acetogenic bacteria such as *Blautia hydrogenotrophica* may not have much of an effect on the production of acetate, they are essential for the digestive tract’s gas disposal process, which supports intestinal health [23].

It has been reported that the following gut microbiota are involved in the production of propionate SCFAs: *Bacteroidetes* and *Negativicutes*, *Megasphaera elsdenii*, *Lachnospiraceae*, and *Coprococcus catus*. Bacteroidetes and Negativicutes utilize vitamin B12-dependent pathways to convert succinate to propionate, which is an essential process in their metabolic activities [25,26]. On the other hand, other bacteria, such as *Megasphaera elsdenii*, *Lachnospiraceae*, and *Coprococcus catus* can produce propionate from lactate using pathways involving succinate or acrylate [27,28].

Lastly, the following gut microbiota species are involved in the production of butyrate: Faecalibacterium prausnitzii, a member of the Clostridium leptum group (clostridial cluster IV), Roseburia species, and members of the Clostridium coccoides group (clostridial cluster XIVa). Their ability to ferment dietary fibers and produce butyrate contributes to the overall balance of the gut microbiota [29,30]. This diversity in metabolic pathways highlights how different bacterial groups contribute to SCFA production in the gut, influencing host health and metabolism.

## 6. An Overview of the Roles of Short-Chain Fatty Acids in Health and Disease

Short-chain fatty acids have several health benefits, including supporting colon health by nourishing colon cells and maintaining intestinal barrier integrity, regulating appetite and promoting satiety, modulating the immune system and reducing inflammation, regulating the pH, and contributing to metabolic health and insulin sensitivity [31]. The biological effects of SCFAs on the host, including their anti-obesity, anti-inflammatory, immunoregulatory, anti-diabetic, cardiovascular, hepatoprotective, and neuroprotective properties, help to delay the progression of disease [32].

### 6.1. The Role of the Three Common SCFAs (Propionate, Butyrate, and Acetate) in Health and Disease

#### 6.1.1. Acetate

Acetate is a precursor for the synthesis of cholesterol and fatty acids and can be used as an energy source in various tissues [33]. Acetate directly suppresses appetite in the central nervous system, especially in the hypothalamus [34]. It is the first and most abundant SCFA in portal circulation, and it also appears to be a ligand for G protein-coupled receptors GPR 41 and GPR 43 [31]. GPR 41 and 43 are also known as free fatty acid receptors (FFARs) 3 and 2 [31]. The activation of these receptors could be responsible for some of the gut microbiome’s physiological effects. According to reports, GPR41 regulates energy in response to the SCFAs generated by the gut microbiota [35], while GPR43 regulates the body’s energy consumption, while maintaining metabolic balance by acting as a sensor for excessive dietary energy [35]. Through its effects on lipid metabolism and glucose homeostasis, acetate regulates body weight and insulin sensitivity in both humans and animals, preserves energy balance and metabolic homeostasis, resists oxidation and mitochondrial stress, and boosts immunity [36]. Data from both humans and animals showed that acetate has a positive impact on host energy and substrate metabolism by secreting gut hormones such as peptide YY (PYY) and glucagon-like peptide-1 (GLP-1) [37]. This in turn influences appetite by lowering systemic pro-inflammatory cytokine levels, reducing whole-body lipolysis, and increasing energy expenditure and fat oxidation [37]. Additionally, acetate helps to keep the pH of the colon acidic, promoting the growth of beneficial bacteria while inhibiting the growth of potentially dangerous pathogens [31].

#### 6.1.2. Propionate

Propionate is the second most abundant SCFA, similar to acetate in terms of its prevalence. It has effects on various metabolic processes, including gluconeogenesis (the production of glucose from non-carbohydrate sources) and lipid metabolism [31]. It has also been demonstrated that propionate affects satiety and hunger [38]. However, the precise mechanism remains unclear [38]. Propionate has been demonstrated to be the key ligand for GPR 41 and GPR 45. It increases leptin levels, the primary hormone involved in the regulation of hunger and satiety in the adipocytes, which results in a decrease in energy intake [18]. Propionate increases the secretion of GLP-1by intestinal epithelial cells (IECs) in humans, resulting in an inhibition of hepatocyte adipogenesis, and decreased fat deposition [36]. Additionally, propionate contributes to hepatic gluconeogenesis by inhibiting the liver’s ability to synthesize cholesterol through its action on 3-hydroxy-3-methyl-glutaryl-CoA (HMG) reductase [36].

#### 6.1.3. Butyrate

Butyrate is the least abundant SCFA, despite being produced in large amounts, as compared to acetate and propionate [39]. It helps maintain the intestinal health of the colon and provides essential energy to colonic epithelial cells [39]. It has been shown that butyrate can restore the epithelial barrier of the airway by reducing the expression of Interleukin-4 (IL-4), zonula occludens protein 1 (ZO-1), and IL-6. This was demonstrated in 16 human bronchial epithelial cells (16HBE) [36]. This causes the phosphorylation of c-Jun N-terminal kinase (JNK) and extracellular signal-regulated protein kinase 1/2 (ERK1/2) [36]. Moreover, it reduces colonic inflammation and limits the occurrence and progression of colon cancer, primarily by regulating cell proliferation and apoptosis and improving intestinal barrier integrity [36]. Normal IECs derive their energy from butyrate, but malignant colon cells use glucose as their primary energy source because of the Warburg effect [36]. Figure 3 illustrates the benefits of SCFAs in health and diseases.

## 7. The Benefits of SCFAs in T2DM

Numerous studies have shown the beneficial roles of SCFAs in patients with T2DM. According to Zhang et al. (2021), SCFAs can influence insulin sensitivity and glucose metabolism in relation to the onset of diabetes via a variety of pathways. These pathways include (i) increasing intestinal gluconeogenesis, (ii) increasing energy supply, (iii) improving intestinal gluconeogenesis, (iv) preserving the integrity of the intestinal barrier, (v) preserving the anaerobic environment of the intestine, and (vi) increasing immunity [40]. To be more specific, acetate SCFAs are energy-regulated signaling molecules that can bind to any free fatty acid receptor 2 or 3 (FFAR2 or FFAR3) on the surface of intestinal L cells [27]. When colon L cells are stimulated further, intestinal hormones such as PYY and GLP-1 are released [28]. These hormones are speculated to slow stomach emptying by lowering glucagon levels, suppressing appetite, and promoting insulin release [28]. Among SCFAs, butyrate is the main energy source, accounting for 5–10% of the colon’s total energy consumption [41]. It stimulates intestinal gluconeogenesis by activating the expression of genes involved in intestinal gluconeogenesis in a Cyclic AMP (cAMP)-dependent manner. Interestingly, propionate functions similarly to butyrate, as a substrate for gluconeogenesis, which is known to activate genes associated with intestinal gluconeogenesis via the intestine–brain nerve circuit, including FFAR3, which controls blood sugar and lipid metabolism [42].

Importantly, maintaining the intestinal barrier’s integrity is critical because pro-inflammatory chemicals such as lipopolysaccharide (LPS) weaken the intestinal barrier in people with T2DM, resulting in insulin resistance [43]. A high-fat diet might be associated with increased gut permeability, allowing LPS to enter the portal system and reach the bloodstream [44]. Once LPS enters the circulation through the portal system, it reaches the liver, where it can contribute to hepatic inflammation and insulin resistance [43]. In summary, the presence of LPS in the blood (metabolic endotoxemia) can cause insulin resistance by inducing inflammatory reactions. Notably, changes in diet can disrupt the balance of microorganisms in the gut. This disruption can decrease the production of proteins that keep the gut lining intact, leading to a leakier gut, where substances can pass through more easily [43]. Studies have indicated that butyrate modulates the production and functionality of several proteins that are essential for maintaining the intestinal barrier’s integrity. Butyrate enhances the expression of Claudin-1 by interacting with the transcription factor SP1 and modifying the promoter region of the Claudin-1 gene. This increased expression of Claudin-1 leads to the redistribution of ZO-1 to the cell membrane, where it helps form tight junctions, thereby strengthening the barrier function of epithelial cells [44,45]. Additionally, butyrate helps maintain an anaerobic gut environment by influencing the metabolism of intestinal cells and inhibiting the growth of harmful bacteria. This preservation of gut health and microbiota balance plays a significant role in improving insulin resistance and protecting against obesity and type 2 diabetes [40,46].

Considering the fact that T2DM patients have ongoing low-grade inflammation, having high levels of SCFAs may be beneficial to these patients because SCFAs play a role in increasing immunity [47]. Butyrate has been shown to have an anti-inflammatory effect by promoting the development of regulatory T cells (Treg) and lowering inflammation, which boosts immunity [48]. In this context, SCFAs may be a viable strategy for preventing diabetes by modulating metabolic response. However, due to a lack of studies focusing on the relationship between the effect of SCFAs and metabolic response, there is no clear conclusive evidence to support their therapeutic benefits on glucose homeostasis in humans [37]. As a result, additional well-controlled long-term intervention studies are needed to validate SCFAs’ beneficial role in metabolic diseases [23]. The beneficial impacts of SCFAs on glucose metabolism, lipid metabolism, and the immune system in diabetics are demonstrated in Figure 4.

## 8. Methods Used for Quantification of SCFAs

To fully understand the important role that SCFAs play in host health, it is crucial to be able to quantify them in a variety of biological samples [40]. Several methods have been employed to analyze SCFAs in biological samples. These methods include gas chromatography (GC) with flame ionization detection (FID) [49], gas chromatography–mass spectrometry (GC/MS) [50], high-performance liquid chromatography (HPLC) coupled with ultraviolet (UV) detection [51], electrochemical detection (ECD) [52]. Furthermore, SCFA measurement has also been performed using capillary electrophoresis (CE). Among these methods, gas chromatography–mass spectrometry (GC/MS) has been shown to be the most accurate and cost-effective technique for the quantification of SCFAs in human plasma, serum, and feces [50].

### 8.1. HPLC Coupled with Ultraviolet (UV) Detector, Electrochemical Detector (ECD), and Mass Spectrometry (MS) Used for SCFAs Quantification

HPLC is a powerful analytical technique based on the principles of liquid chromatography, using high pressure to force the solvent through a column packed with the stationary phase, resulting in the separation of the components of a mixture. The choice of stationary phase, mobile phase, and operational parameters can be optimized for specific types of analyses [51,53]. When HPLC is combined with various detection methods, it offers versatile options for quantifying SCFAs. Three commonly used detection methods coupled with HPLC for the quantification of SCFAs include (i) ultraviolet (UV) detector, (ii) electrochemical detector (ECD), and (iii) mass spectrometry (MS) detector [53].
(i)HPLC coupled with ultraviolet (UV) detector (HPLC-UV)

The HPLC-UV detector relies on the absorption of UV light by the analytes. The extent of absorption at a specific wavelength correlates with the concentration of the analyte [53,54]. The benefits of HPLC-UV include being more cost-effective, easier to use, and suitable for routine analysis with moderate sensitivity requirements. The detection limits for HPLC-UV are often greater than those for GC-MS, which means that this detector may not be as sensitive for detecting low amounts of SCFAs. Furthermore, this type of detector has lower resolution compared to GC-MS, potentially leading to less accurate quantification if the SCFAs are not well separated [51,53,54].
(ii)HPLC coupled with an electrochemical detector (HPLC-ECD)

This type of detector measures the current resulting from the oxidation or reduction of analytes at an electrode surface. The analyte’s concentration determines how much current flows [52,55]. The HPLC-ECD method is highly sensitive and selective, making it suitable for detecting low concentrations of SCFAs. When comparing HPLC-ECD with GC-MS for SCFA quantification, one challenge with HPLC-ECD is its sensitivity, as it can optimize specific SCFAs but not all of them [52,53].

(iii)HPLC coupled with mass spectrometry (MS) detector (LC-MS)

The LC-MS is an analytical technique that separates ionized particles such as atoms, molecules, and clusters based on changes in charge-to-mass ratios (mass/charge; m/z) and can be used to calculate the molecular weight of the particles. The MS technique can identify, measure, and provide structural information about SCFAs due to its high sensitivity and specificity. However, LC-MS is more expensive to purchase and maintain than GC-MS [52].

### 8.2. Capillary Electrophoresis (CE) for Quantification of SCFAs

Capillary electrophoresis (CE) separates charged molecules within a capillary when subjected to an electric field. The separation relies on the electrophoretic mobility of ions through the gel, which depends on the molecules’ mass, shape, and charge [56]. CE is advantageous due to its minimal sample pretreatment, making it fast and convenient for routine analyses. However, it has limitations, including low reproducibility and repeatability, and it requires a higher analyte concentration due to the small quantity of the injected sample [53].

### 8.3. Gas Chromatography (GC) Coupled with Flame Ionization Detector (FID) for Quantification of SCFAs

Gas chromatography (GC) is a powerful analytical technique used to separate, identify, and quantify the compounds in a mixture. The basic principle of GC involves volatilizing the sample, transporting it through a column by an inert carrier gas, separating the components based on their interactions with the stationary phase, detecting the separated components, and analyzing the data to identify and quantify the components [53]. Several detectors are commonly employed to quantify short-chain fatty acids (SCFAs) using gas chromatography (GC), based on their sensitivity and suitability for the target analytes [53]. Common GC detectors for SCFA quantification include flame ionization detectors (FIDs) and mass spectrometry (MS). FIDs measure ions formed during the combustion of organic compounds in a hydrogen flame. For SCFA analysis, the FID method is the most commonly used detector due to its sensitivity and ease of use with organic acids. However, when higher sensitivity and specificity are required, especially in complex biological samples, MS detectors are preferred [52,53].

### 8.4. Gas Chromatography–Mass Spectrometry as a Strategy to Improve the Quantification of Short-Chain Fatty Acids in LMICs

For the quantification of SCFAs using GC/MS, pre-column derivatization is frequently necessary. Fatty acids are often derivatized into their methyl ester or trimethylsilyl ester derivatives because GC/MS requires volatile compounds [50]. Importantly, signal overlap may occur because of SCFAs’ low intrinsic boiling point, which is comparable to that of derivatization agents that are commonly used. For this reason, the derivatization agent must be carefully chosen [50]. Furthermore, quite a few derivatization agents are sensitive to moisture, which makes them unsuitable for the aqueous matrices that contain SCFAs. It has been reported that these limitations have been overcome by using isobutyl chloroformate and pentafluorobenzyl bromide (PFBBr) derivatization reagents [50]. The lengthy derivatization process of the samples can cause deviations in analysis because of the evaporation that occurs during sample preparation. As an alternative to the derivatization of SCFAs, direct aqueous injection of a biological sample can be used [40]; however, the complex biological components can contaminate the GC/MS system. Consequently, a better technique that lowers contamination of the GC/MS is needed. Lastly, sample extraction for this method is a labor- and time-intensive process with a high loss rate of analyte molecules in GC/MS analysis [56]. Therefore, it is imperative to validate alternative extraction methods that have the potential to improve sample recovery and are faster and more efficient. This could therefore aid in improving the ability to quantify SCFAs in people who are at risk of developing T2DM.

Despite these limitations, GC/MS techniques are generally preferred over other chromatographic methods when it comes to quantifying SCFAs [50]. The reason is that they have high sensitivity and specificity, which allows them to detect small concentrations of analytes in biological samples such as feces [57]. They also offer high resolution, which makes it easier to identify and separate closely related compounds. The instrument can quantify various compounds present in a sample, often over a wide concentration range, and also provides high resolution [58]. Finally, the costs involved in the running of GC/MS instrumentation are lower than those of LC/MS because GC/MS requires less specialized operator training and has fewer parts that need to be replaced or maintained on a regular basis [57]. As a result, by implementing this strategy into practice, especially in poor countries (LMICs) with limited laboratory resources and a high prevalence of T2DM, the communities in the LMICs will be better informed about the advantageous link between SCFAs and T2DM, and ultimately more lives will be saved.

## 9. Conclusions

Currently, the prevalence of T2DM is increasing at a rapid pace, especially in LMICs, which has caused concerns in the global healthcare communities because over one-third of diabetes-related deaths involve individuals under the age of 60 years. Previous studies have indicated that consuming highly fermentable dietary fibers, which produce SCFAs, may benefit people with T2DM and those who are susceptible to the disease by improving their immune systems, lipid and glucose metabolism, and overall immune system function. The lack of research reporting on the advantageous link between higher levels of SCFAs and glucose metabolism, however, has resulted in contradictions regarding the therapeutic benefits of SCFAs on human glucose homeostasis. By addressing this knowledge gap, we can potentially unlock new insights into how dietary fibers and SCFAs can contribute to the overall health, particularly for individuals with metabolic disorders such as T2DM. Hence, more research in this area, pertaining to human populations, is desperately needed.

This paper highlights the importance of encouraging people to consume sufficient quantities of fiber-rich foods to reduce the risk of developing T2DM. This review paper further highlights the importance of implementing the GC/MS method to quantify SCFAs because of the method’s specificity and cost-effectiveness for poor countries with resource-constrained settings. The adoption and use of this analytical system may improve awareness of the advantageous link between SCFAs and glucose metabolism in LMICs, particularly for those individuals who are more susceptible to T2DM. This allows for the early implementation of therapeutic and preventative measures aimed at reducing the high prevalence of T2DM cases and the disease’s death toll.

## 10. Recommendation for Future Studies

For the future, the aim is to validate accurate, fast, and cheaper alternative methods for sample extraction that could improve sample recovery and reduce the turnaround time for SCFA quantification in GC/MS analysis. It is necessary to determine the levels of SCFAs in patients with T2DM to know the concentration and the type of SCFAs present. Measuring them will enable clinicians to accurately advise and recommend the amount of dietary fiber a patient needs/requires.

## Figures and Tables

**Figure 1 diagnostics-14-01636-f001:**
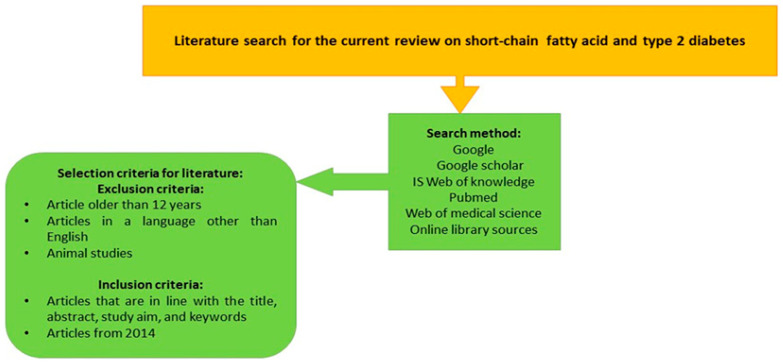
The flow diagram for the literature search conducted in the current narrative review paper.

**Figure 2 diagnostics-14-01636-f002:**
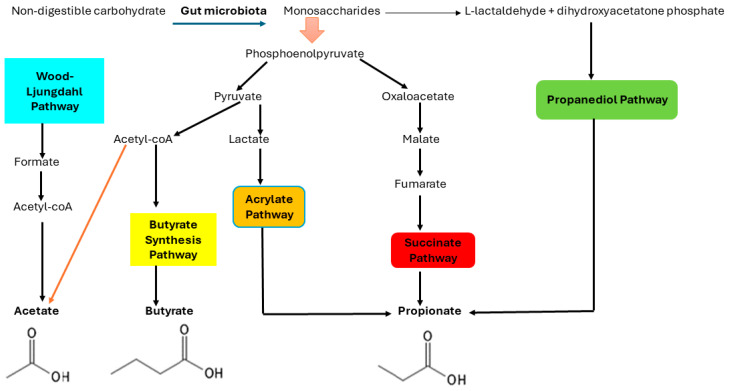
A schematic overview of the pathways involved in the fermentation of dietary fiber and carbohydrates by anaerobic gut bacteria to produce acetate, propionate, and butyrate. Acetate can be synthesized directly from acetyl CoA or through the Wood–Ljungdahl pathway using formate. Propionate can be synthesized from phosphoenolpyruvate (PEP) via the succinate decarboxylation pathway, the acrylate pathway, which reduces lactate to propionate, and the propanediol pathway. Butyrate is produced by the enzyme butyrate-kinase from the condensation of two molecules of acetyl CoA, which results in butyrate, or by utilizing exogenously derived acetate via the enzyme butyryl CoA/acetate-CoA-transferase. The figure depicts the biochemical structure of the most common SCFAs [acetate (C2), butyrate (C4), and propionate (C3)] (The author created the figure in PowerPoint Version 2406).

**Figure 3 diagnostics-14-01636-f003:**
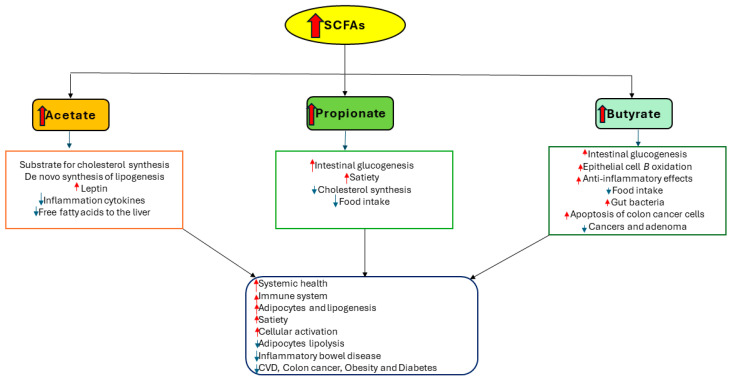
The three most common SCFAs’ functions in health and diseases. The red arrows indicate an increase, while the blue arrows signify a decrease. (The author created the figure in PowerPoint Version 2406).

**Figure 4 diagnostics-14-01636-f004:**
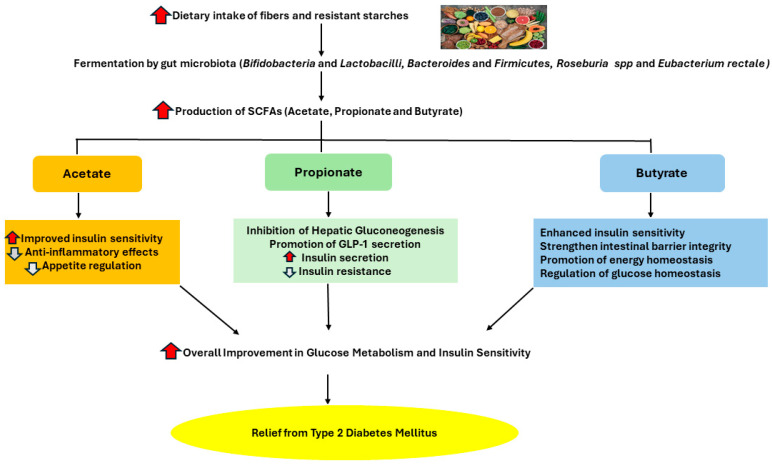
Summary of the beneficial effects of short-chain fatty acids on glucose metabolism, lipid metabolism, and immune regulation in individuals with T2DM. The red arrows indicate an increase, while the white arrows signify a decrease. (The author created the figure in PowerPoint Version 2406).

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
