# Peer review of "Type 2 Diabetes Mellitus in Low- and Middle-Income Countries: The Significant Impact of Short-Chain Fatty Acids and Their Quantification"

_diagnostics, 2024, doi:10.3390/diagnostics14151636_

Round 1

Reviewer 1 Report

Comments and Suggestions for Authors

This manuscript deals with an important topic in T2DM, which is the impact of short chain fatty acids. The manuscript is well prepared in an ordered manner and supported by four figures. It should be accepted. The first sentence of "Introduction" is a good description of T2DM. However, It is recommended to show a list of abbreviations and a final conclusion in the manuscript.

Comments on the Quality of English Language

It is fine.

Reviewer 2 Report

Comments and Suggestions for Authors

The flow diagram should present the mechanism of action of short chain fatty acids to relieve the T2DM and microbiota producing short chain fatty acids from different nutrients.

Please increase the resolution of all figures.

I suggest to add the literature about major gut microbiota who are responsible for the production of short-chain fatty acids.

It is also suggested to display in detail how short-chain fatty acids such as Acetate, Propionate, Butyrate inflict their beneficial impact on T2DM.

Please discuss in detail all methods used for quantification of SCFAs and write their advantages and draw backs. 

Comments on the Quality of English Language

Just required proofreading

Reviewer 3 Report

Comments and Suggestions for Authors

The review by Kumato et al focuses on the role of short-chain fatty acids (SCAFs) formed from dietary fibers by intestinal bacteria in preventing metabolic disorders manifested in the development of type 2 diabetes mellitus (T2DM) associated with insulin resistance. The authors, cite a number of data indicating that the risk of T2DM may be reduced in individuals who consume a dietary intake of coarse dietary fibers. Parallel studies support the formation of short-chain fatty acids by intestinal microorganisms from fiber-rich foods. The impaired synthesis of short-chain fatty acids in T2DM patients is caused by pathogenic bacteria unable to synthesise SCAFs.

The positive role of short-chain fatty acids with unbranched chains (acetate, propionate, butyrate) and the pathogenic role of branched fatty acids is separately noted.

The review summarises the scheme of the main metabolic pathways leading to the formation of SCAFs as a result of the activity of intestinal bacteria; the intracellular mechanisms mediating the involvement of acetate, propionate and butyrate in physiological processes positive for the organism, leading to normalization of inflammatory and immune mechanisms, cellular activity, lipid metabolism and reduction of the risk of diabetes and a number of other diseases are considered separately. Cell receptors and protein-mediators involved in the realization of positive effects of SCAFs are considered specifically.

The review concludes by analyzing the feasibility of using modern methods of laboratory determination of fatty acids in countries with low-income populations. The usefulness of allocating funds in low-middle income countries for such studies to reduce mortality among the population is pointed out.

The review makes a good impression with its combination of thoroughness and brevity.

The reviewer has no comments of principle to the authors.

Minor comments.

1. Lines 93 and 107: Caecum or cecum?!

2. It is desirable to introduce the notation “PYY” and “GLP-1” at the first mention of “Peptide YY” and “Glucagon-like peptide-1” (line 174), and then to give them in the text as symbols (lines 218-219 et seq.).

3. 3-hydroxy-3-methyl-glutaryl-CoA (HMG) reductase” (lines 190-191) as well as “Cyclic AMP (cAMP)-dependent manner” (line224) is more accurate.

Round 2

Reviewer 2 Report

Comments and Suggestions for Authors

The authors have incorporated all suggestions. Therefore manuscript is recommended for publication.